# Isogeometric Analysis of Graphene-Reinforced Functionally Gradient Piezoelectric Plates Resting on Winkler Elastic Foundations

**DOI:** 10.3390/ma15165727

**Published:** 2022-08-19

**Authors:** Yanan Liang, Shijie Zheng, Dejin Chen

**Affiliations:** 1State Key Laboratory of Mechanics and Control of Mechanical Structures, Nanjing University of Aeronautics and Astronautics, Nanjing 210016, China; 2Liaoning Hongyanhe Nuclear Power Co., Ltd., Dalian 116001, China

**Keywords:** graphene-reinforced piezoelectric composite plate, functionally graded material, isogeometric analysis, refined plate theory, static elastic deflection, free vibration, buckling behavior, Winkler elastic foundations

## Abstract

In this paper, the refined plate theory (RPT), Hamilton’s principle, and isogeometric analysis (IGA) are applied to investigate the static bending, free vibration and buckling behaviors of functionally graded graphene-platelet-reinforced piezoelectric (FG-GRP) plates resting on a Winkler elastic foundation. The graphene platelets (GPLs) are distributed in polyvinylidene fluoride (PVDF) as a power function along the plate thickness direction to generate functionally gradient materials (FGMs). The modified Halpin–Tsai parallel model predicts the effective Young’s modulus of each graphene-reinforced piezoelectric composite plate layer, and the rule of the mixture can be used to calculate the effective Poisson’s ratio, mass density, and piezoelectric properties. Under different graphene distribution patterns and boundary conditions, the effects of a plate’s geometric dimensions, GPLs’ physical properties, GPLs’ geometric properties and the elastic coefficient of the Winkler elastic foundation on deflections, frequencies and bucking loads of the FG-GRP plates are investigated in depth. The convergence and computational efficiency of the present IGA are confirmed versus other studies. Furthermore, the results illustrate that a small amount of GPL reinforcements can improve the FG-GRP plates’ mechanical properties, i.e., GPLs can improve the system’s vibration and stability characteristics. The more GPL reinforcements spread into the surface layers, the more effective it is at enhancing the system’s stiffness.

## 1. Introduction

Functionally gradient materials (FGMs), first developed in 1987, are composite materials with two or more distinct phases. FGMs provide designers with a range of options to distribute strength and stiffness in an appropriate manner [1]. Since FGMs have great properties and match well with many engineering challenges, many researchers have focused on FG structures formed of FGMs. Additionally, their mechanical properties, such as bending, vibration, and buckling behaviors, have been explored and realized more extensively [2,3,4]. Novoselov et al. [5] succeeded in extracting graphene platelets (GPLs) in 2004. Since then, a number of theoretical and experimental studies have focused on understanding the behavior of GPLs. Because GPLs have high strength, stiffness as well as good thermal and electrical properties [6,7], many experimental and numerical results have shown that combining GPL reinforcements with FGMs can improve the mechanical, piezoelectric, and thermoelectric properties of the composite structures [8,9,10]. According to Lee et al. [11], when the graphene volume fraction reached 1.6%, the strength of the composites increased to roughly 80% compared with pure epoxy. As a result, functionally graded GPL-reinforced composites (FG-GPLRCs) have become one of the most popular composites today, with applications in a wide range of industries. In the field of FG-GPLRCs, Feng et al. [12,13] took advantage of the Timoshenko beam theory to investigate the mechanical properties related to static analysis and free vibration of GPL-reinforced multilayer polymer nanocomposite beam under different geometrical properties and distribution patterns of GPLs. Yang et al. [14] proposed an analytical solution for the nonlinear buckling of FG-GPLRC arches subjected to a central point load. Based on the nonlocal strain gradient theory, Sahmani et al. [15] examined the nonlinear bending behaviors of functionally graded porous beams reinforced with graphene platelets. The authors of [16,17,18,19,20] investigated the static and kinetic properties of GPL-reinforced functionally gradient multilayer nanocomposites. Liu et al. [21] adopted the state-space method and Euler–Bernoulli beam theory to calculate the fundamental frequency and critical buckling loads of axially functionally graded graphene-reinforced nanocomposite beams. On basis of modified couple stress theory, Timoshenko beam theory and IGA, Li et al. [22] investigated the scale-dependent static bending and free vibration of bi-directional functionally graded graphene-nanoplatelet-reinforced composite microbeams. The influences of power-law indexes, GPL weight fraction, scale parameter and boundary conditions on static bending and free vibration performances of bi-directional functionally graded graphene-nanoplatelet-reinforced composite microbeams are investigated. Recently, Zhao et al. [23] presented a comprehensive review on the state of the art of FG-GPLRC structures, and highlighted future research directions and key technical challenges.

On the other hand, piezoelectric ceramic, one kind of smart material, has excellent electro-mechanical coupling behaviors. Piezoelectric sensors can convert mechanical energy to electrical energy, making them beneficial in civil engineering, automotive and aerospace [24]. With the diversification and complexity of the forms of piezoelectric smart materials, investigating their free vibration, buckling behaviors, shape control and other related problems is becoming more and more significant [25,26,27]. Layek et al. [28] discovered that adding 0.75% weight fraction of GPLs to a polyvinylidene fluoride (PVDF) increased the storage modulus and Young’s modulus of composites by 124% and 121%, respectively. Therefore, combining piezoelectric materials with the GPL reinforcements creates new smart piezoelectric composite materials with superior characteristics, and widely applying them in all aspects would significantly improve quality of life. In the field of functionally graded GPL-reinforced piezoelectric (FG-GRP) composites, Abolhasani et al. [29] effectively prepared GNPL-reinforced PVDF composite nanofibers and experimentally examined their crystallinity, polymorphism, and electrical output for the first time. Based on the differential quadrature method (DQM), Mao and his group [30] present analytical investigations on the linear and nonlinear vibration as well as buckling and postbuckling behaviors [31] of FG-GRP plates, and they used the extended Eringen’s nonlocal elasticity theory to study the scale effect on the linear and nonlinear vibrations [32]. The vibrational characteristics of the graphene-reinforced cylindrical nanoshell coupled with a piezoelectric actuator are studied by Ghabussi et al. [33]. Yang et al. [34] investigated the nonlinear buckling behavior of graphene-reinforced dielectric composite arches under applied electric and uniform radial load. Fu et al. [35] presented an analytical solution for typical static problems of the graphene-reinforced-composite-laminated beams integrated with piezoelectric macro-fiber-composite actuators. In conclusion, it has been discovered that a small amount of graphene reinforcements dispersed into the polymer matrix can significantly improve stiffness and has a positive effect on the structure’s stability, i.e., GPLs will have a significant effect on the static and dynamic behaviors for the piezoelectric composite plate.

Many scholars have proposed theoretical models that combine shear deformation theory [36] with various numerical analytic methods [37,38,39] to efficiently explore the electrical and mechanical properties of piezoelectric composites. It should be noted that the classical plate theory (CPT) neglects the transverse shear deformation; hence, CPT is only applied to analyze thin plates. The first-order shear deformation theory (FSDT) treats the shear stress along the thickness direction as a constant, resulting in poor analytical accuracy. The higher-order shear deformation theory (HSDT) is independent of shear correction factor by introducing higher-order terms in the displacement field approximation. However, HSDT is computationally expensive. Relatively speaking, the refined plate theory (RPT) [40] offers apparent advantages. As a simplification of the third-order shear deformation theory (TSDT), RPT not only preserves the benefits of TSDT, but also eliminates an unknown variable relative to TSDT. On the other hand, RPT must match C^1^-continuity of the displacement fields, which presents challenges to the traditional finite element analysis (FEA) framework [41]. Furthermore, the generally employed numerical methods would result in disagreement between the geometric model and the analytical model when the geometries are complicated, unavoidably introducing a significant number of interactions between computer aided design (CAD) and computer aided engineering (CAE) data. To address such issues, Hughes et al. [42] originally presented the notion of isogeometric analysis (IGA) in 2005, which is a novelty computational approach that smoothly integrates CAD with CAE. The fundamental concept of IGA is to apply spline functions (non-uniform rational B-splines [43,44], T-splines and so on) to express the geometry and approximate the unknown fields. IGA not only avoids the approximation error caused by the interpolation function of polynomials, but it also solves the C^1^-continuity of displacement fields required by RPT. In recent years, researchers have frequently used IGA to analytically calculate the mechanical properties of functionally graded composites. On the basis of IGA and TSDT, Kiani et al. [45,46] investigated the large amplitude free vibration and post-buckling properties of GPL-reinforced laminates in a thermal environment. In the context of FSDT and HSDT, Li et al. [47] utilized IGA to investigate the static linear bending, natural frequency, and buckling behavior of GPL-reinforced FG porous plates. Liu et al. [48] combined IGA with simple first-order shear deformation theory (S-FSDT) to analyze the static bending, free vibration, and dynamic response of FG plates integrated with piezoelectric actuators and sensors. Phung-Van et al. [49] investigated the static bending, free vibration, and dynamic control of composite plates integrated with piezoelectric actuators and sensors based on the combination of HSDT and IGA.

To the best of the authors’ knowledge, no literature is available for the analysis of functionally graded graphene-platelet-reinforced piezoelectric structures based on isogeometric analysis. As a result, the purpose of this work is to investigate the static bending, free vibration and buckling behaviors of functionally graded graphene-platelet-reinforced piezoelectric plates embedded on a Winkler elastic foundation using isogeometric analysis and refined plate theory. The object of this paper is to development isogeometric analysis of graphene-reinforced functionally gradient piezoelectric plates resting on Winkler elastic foundations. This paper is organized as follows. In Section 2, the graphene-reinforced functional gradient piezoelectric plate model is established by refined plate theory and Hamilton’s principle. Isogeometric analysis formulations are established in Section 3. In Section 4, detailed parameter investigations are discussed to demonstrate the influences of a plate’s geometric dimensions, GPLs’ geometric properties, GPLs’ volume fraction and the elastic coefficient of the Winkler elastic foundation on mechanical behaviors. Finally, some main conclusions are drawn in Section 5.

## 2. Theoretical Formulation

### 2.1. Material Properties

As illustrated in Figure 1, an FG-GRP plate of length *a*, width *b*, and thickness *h* on Winkler elastic foundation is established. The FG-GRP plate is supposed to be made of a multi-layer polymer composite reinforced by graphene platelets, where the GPL reinforcements are functionally graded along the thickness direction with a power law distribution. Furthermore, the thickness of each independent layer may be described as Δh=h/N.

In this research, three graphene distribution patterns are generated by smoothly varying the graphene volume fractions along the thickness direction, as shown in Figure 2: denoted as the U type, the X type, and the O type. The graphene volume fraction of the top and bottom surfaces is significantly larger than that of the mid-plane in the X type, indicating that the reinforcing effect of GPL in the top and bottom surfaces is more remarkable than that in the mid-plane. On the contrary, the graphene volume fraction of the top and bottom surfaces is significantly lower than that in the mid-plane in the O type, indicating that GPL reinforcements have opposite strengthening effects for both the O type and X type. The graphene volume fraction maintains constant along the thickness direction for the U type, resulting in a uniform distribution on the whole. Both the X and O types are functionally gradient distributions in which the graphene volume fraction increases or decreases linearly and symmetrically from the mid-plane to the top and bottom surfaces of the composite.

The total graphene volume fraction is expressed as $V_{gpl}$, and that the minimum graphene volume fraction in each layer is expressed as $V^{*}$ for the X and O types, then
(1)$$V^{*}=\frac{2}{1+N/2}V_{gpl}$$

The volume fraction in the *k*-th layer of the FG-GRP plate can be written as follows:


The U type:(2)$$V_{k}=V_{gpl}$$The X type:(3)$$\begin{array}{l} V_{k}=\left(\frac{N}{2}+1-k\right)V^{*},\;\mathrm{when}\;k\leq \frac{N}{2}\\ V_{k}=\left(k-\frac{N}{2}\right)V^{*},\;\mathrm{when}\;k>\frac{N}{2} \end{array}$$The O type:(4)$$\begin{array}{l} V_{k}=kV^{*},\;\mathrm{when}\;k\leq \frac{N}{2}\\ V_{k}=\left(N+1-k\right)V^{*},\;\mathrm{when}\;k>\frac{N}{2} \end{array}$$


As demonstrated by Layek et al. [28], GPL reinforcements are more easily dispersed in the matrix material, as the graphene volume fraction is less than 1%. As a result, the FG-GRP plate with a graphene volume fraction of less than 1% is the subject of this research. The effective Young’s modulus of an FG-GRP plate is given for the *k*-th layer as
(5)$$E_{k}=\frac{1+\frac{2a_{gpl}}{3h_{gpl}}\eta_{L}V_{k}}{1-\eta_{L}V_{k}}E_{M}$$
where
(6)$$\eta_{L}=\frac{\frac{E_{G}}{E_{M}}-1}{\frac{E_{G}}{E_{M}}+\frac{2a_{gpl}}{3h_{gpl}}}$$
in which the subscripts *G* and *M* symbolize graphene platelet and matrix material, and $a_{gpl}$, $b_{gpl}$ and $h_{gpl}$ represent the GPL’s length, width and thickness, respectively.

Furthermore, the effective material properties, such as Poisson’s ratio, density, piezoelectric constant, and dielectric constant for the *k*-th layer may be predicted by
(7)$$\begin{array}{l} \rho_{k}=\rho_{G}V_{k}+\rho_{M}\left(1-V_{k}\right)\\ \nu_{k}=\nu_{G}V_{k}+\nu_{M}\left(1-V_{k}\right)\\ e_{km,k}=e_{km,G}V_{k}+e_{km,M}\left(1-V_{k}\right)\\ \kappa_{km,k}=\kappa_{km,G}V_{k}+\kappa_{km,M}\left(1-V_{k}\right) \end{array}$$

### 2.2. Governing Equations for the Refined Plate Theory

The displacement fields (u,v,w) consisting of four unknown variables, according to the refined plate theory, may be described as
(8)$$\begin{array}{l} u\left(x,y,z\right)=u_{0}\left(x,y\right)+zw_{b,x}\left(x,y\right)+f\left(z\right)w_{s,x}\left(x,y\right)\\ v\left(x,y,z\right)=v_{0}\left(x,y\right)-zw_{b,y}\left(x,y\right)+f\left(z\right)w_{s,y}\left(x,y\right)\\ w\left(x,y,z\right)=w_{b}\left(x,y\right)+w_{s}\left(x,y\right) \end{array}$$
where $u_{0}$ and $v_{0}$ represent the in-plane displacement components in the mid-plane, $w_{b}$ and $w_{s}$ denote the transverse bending and shear displacement components, and $f\left(z\right)=-4z^{3}/3h^{2}$.

The in-plane strain and transverse shear strains corresponding to the displacement fields are shown as
(9)$$\begin{array}{l} \varepsilon =\left\{\begin{array}{c} \varepsilon_{xx}\\ \varepsilon_{yy}\\ \varepsilon_{xy} \end{array}\right\}=\left[\begin{array}{c} u_{0,x}\\ v_{0,y}\\ u_{0,y}+v_{0,x} \end{array}\right]+z\left[\begin{array}{c} \;-w_{b,xx}\\ \;-w_{b,yy}\\ \;-2w_{b,xy} \end{array}\right]+f\left(z\right)\left[\begin{array}{c} w_{s,xx}\\ w_{s,yy}\\ 2w_{s,xy} \end{array}\right]=\varepsilon_{m}+z\kappa_{b}+f\left(z\right)\kappa_{s}\\ \gamma =\left\{\begin{array}{c} \gamma_{xz}\\ \gamma_{yz} \end{array}\right\}=\left(1+f^{\prime }\left(z\right)\right)\left[\begin{array}{c} w_{s,x}\\ w_{s,y} \end{array}\right]=\left(1+f^{\prime }\left(z\right)\right)\varepsilon_{s} \end{array}$$

According to the generalized Hooke’s law, the *k*-th layer constitutive relationship is represented as
(10)$$\sigma^{\left(k\right)}={\left\{\begin{array}{c} \sigma_{xx}\\ \sigma_{yy}\\ \tau_{xy}\\ \tau_{xz}\\ \tau_{yz} \end{array}\right\}}^{\left(k\right)}={\left[\begin{array}{ccccc} Q_{11} & Q_{12} & 0 & 0 & 0\\ Q_{21} & Q_{22} & 0 & 0 & 0\\ 0 & 0 & Q_{66} & 0 & 0\\ 0 & 0 & 0 & Q_{55} & 0\\ 0 & 0 & 0 & 0 & Q_{44} \end{array}\right]}^{\left(k\right)}{\left\{\begin{array}{c} \varepsilon_{xx}\\ \varepsilon_{yy}\\ \gamma_{xy}\\ \gamma_{xz}\\ \gamma_{yz} \end{array}\right\}}^{\left(k\right)}$$
where
(11)$$\begin{array}{l} Q_{11}^{\left(k\right)}=Q_{22}^{\left(k\right)}=\frac{E_{k}}{1-\nu_{k}^{2}},\;Q_{12}^{\left(k\right)}=Q_{21}^{\left(k\right)}=\frac{\nu_{k}E_{k}}{1-\nu_{k}^{2}},\\ Q_{66}^{\left(k\right)}=\frac{E_{k}}{2\left(1+\nu_{k}\right)},\;Q_{44}^{\left(k\right)}=Q_{55}^{\left(k\right)}=\frac{E_{k}}{2\left(1+\nu_{k}\right)} \end{array}$$

For the FG-GRP plate, the constitutive equation is expressed as
(12)$$\left[\begin{array}{c} \sigma \\ D \end{array}\right]=\left[\begin{array}{cc} c & -e^{T}\\ e & g \end{array}\right]\left[\begin{array}{c} \bar{\varepsilon }\\ E \end{array}\right]$$
in which σ and $\bar{\varepsilon }$ represent the plate’s stress and strain vectors, D is the electric displacement vector, c denotes the elastic stiffness matrix, e and g are the piezoelectric and dielectric constant matrices, respectively. It should be mentioned that the electric field vector E is given as
(13)$$E={\left\{\begin{array}{ccc} E_{x} & E_{y} & E_{z} \end{array}\right\}}^{T}=-grad\phi =-{\left\{\begin{array}{ccc} 0 & 0 & \frac{\partial \phi }{\partial z} \end{array}\right\}}^{T}$$

According to Equations (9)–(13), the FG-GRP plate’s mid-plane internal force N, bending moment M, high-order bending moment P, and transverse shear force R are written as follows:(14)$$\begin{array}{l} \left\{\begin{array}{c} N\\ M\\ P \end{array}\right\}=D_{m}\left\{\begin{array}{c} \varepsilon_{m}\\ \kappa_{b}\\ \kappa_{s} \end{array}\right\}\;-\left\{\begin{array}{c} N^{E}\\ M^{E}\\ P^{E} \end{array}\right\}\\ R=D_{s}\varepsilon_{s} \end{array}$$
where
(15)$$\begin{array}{l} D_{m}=\left[\begin{array}{ccc} A & B & E\\ B & D & F\\ E & F & H \end{array}\right]\\ {\left(A,B,D,E,F,H\right)}^{\left(k\right)}=\int_{-h/2}^{h/2}\left(1,z,z^{2},f\left(z\right),zf\left(z\right),f^{2}\left(z\right)\right)Q^{\prime }dz\\ {\left(D^{s}\right)}^{\left(k\right)}=\int_{-h/2}^{h/2}{\left(f^{\prime }\left(z\right)+1\right)}^{2}G^{\prime }dz \end{array}$$
and
(16)$$\begin{array}{l} Q^{\prime }=\left[\begin{array}{ccc} Q_{11}^{\left(k\right)} & Q_{12}^{\left(k\right)} & 0\\ Q_{21}^{\left(k\right)} & Q_{22}^{\left(k\right)} & 0\\ 0 & 0 & Q_{66}^{\left(k\right)} \end{array}\right]\\ G^{\prime }=\left[\begin{array}{cc} Q_{44}^{\left(k\right)} & 0\\ 0 & Q_{55}^{\left(k\right)} \end{array}\right] \end{array}$$

The internal force $N^{E}$ and bending moment $M^{E}$, $P^{E}$ induced by an electric field are be defined as
(17)$${\left(N^{E},M^{E},P^{E}\right)}^{\left(k\right)}=\int_{-h/2}^{h/2}e^{T}E\left(1,z,z^{3}\right)dz$$

### 2.3. Weak Form

The Galerkin weak form of the governing equations is calculated using the Hamilton principle [50].
(18)$$\begin{array}{l} \delta \int_{t_{1}}^{t_{2}}Ldt=0\\ L=\int_{\Omega }\left(\frac{1}{2}m{\dot{u}}^{T}\dot{u}-\frac{1}{2}\varepsilon^{T}c\varepsilon +\varepsilon^{T}eE+\frac{1}{2}E^{T}gE\right)d\Omega +\int_{\Omega }\left(u^{T}f_{s}-\phi q_{s}\right)d\Omega +\sum u^{T}F_{p} \end{array}$$
in which u and $\dot{u}$ represent the displacement and velocity vectors, respectively, ϕ denotes electric potential, $f_{s}$ and $F_{p}$ represent surface force and concentrated force, respectively, and $q_{s}$ denotes surface charge.

## 3. Isogeometric Plate Formulation

### 3.1. NURBS Basis Function

The main elements of the B-spline curve are a set of control points $P_{i}$ and a group of knot vectors Ξ formed of non-decreasing parameter values $\xi_{i},\;i=1,\ldots ,n+p$. The definition formula of Ξ is as follows:(19)$$\Xi =\left\{\xi_{1}=0,\cdots ,\xi_{i},\cdots ,\xi_{n+p+1}=1\right\}$$
where $\xi_{i}$, *n* and *p* denote the *i*th knot, the number of basic functions, and the polynomial order, respectively.

According to the Cox–de Boor recursive formula [42], the B-spline basis function is defined as
(20)$$\begin{array}{l} N_{i,0}(\xi )=\{\begin{array}{cc} 1 & \begin{array}{cc} if & \xi_{i}\leq \xi <\xi_{i+1} \end{array}\\ 0 & otherwise \end{array},\begin{array}{cc} \begin{array}{cc} & for \end{array} & p=0 \end{array}\\ N_{i,p}(\xi )=\frac{\xi -\xi_{i}}{\xi_{i+p}-\xi_{i}}N_{i,p-1}(\xi )+\frac{\xi_{i+p+1}-\xi }{\xi_{i+p+1}-\xi_{i+1}}N_{i+1,\;p-1}(\xi ),\begin{array}{cc} \begin{array}{cc} & for \end{array} & p\geq 1 \end{array} \end{array}$$

It should be noted that the ratio 0/0 in the preceding Equation (20) is given as zero.

The B-spline basis function is mostly made up of specified spline basis function orders and related knot vectors, and its expression is expressed as follows:(21)$$C(\xi )=\sum_{i=1}^{n}N_{i,p}(\xi )P_{i}$$

The B-spline function can be a particular case of the NURBS function when weights of control points are constant. The bi-variate NURBS basis functions are written as
(22)$$R_{i,j}^{p,q}(\xi ,\eta )=\frac{N_{i,p}(\xi )M_{j,q}(\eta )w_{i,j}}{\sum_{i=1}^{n}\sum_{j=1}^{m}N_{i,p}(\xi )M_{j,q}(\eta )w_{i,j}}=\frac{N_{i,p}(\xi )M_{i,q}(\eta )w_{i,j}}{W(\xi ,\eta )}$$

The NURBS spline surfaces are defined by supplying knot vectors and control points in two parameter directions, and the derivative of ξ and η in a bivariate NURBS basis function is expressed as
(23)$$S(\xi ,\eta )=\sum_{i=1}^{n}\sum_{j=1}^{m}R_{i,j}^{p,q}(\xi ,\eta )P_{i,j}$$
(24)$$\begin{array}{l} \frac{\partial R_{i,j}^{p,q}(\xi ,\eta )}{\partial \xi }=w_{i,j}\frac{\frac{\partial N_{i,p}(\xi )}{\partial \xi }M_{j,q}(\eta )W(\xi ,\eta )-\frac{\partial W(\xi ,\eta )}{\partial \xi }N_{i,p}(\xi )M_{j,q}(\eta )}{{\left[W(\xi ,\eta )\right]}^{2}}\\ \frac{\partial R_{i,j}^{p,q}(\xi ,\eta )}{\partial \eta }=w_{i,j}\frac{\frac{\partial M_{j,q}(\eta )}{\partial \eta }N_{i,p}(\xi )W(\xi ,\eta )-\frac{\partial W(\xi ,\eta )}{\partial \eta }N_{i,p}(\xi )M_{j,q}(\eta )}{{\left[W(\xi ,\eta )\right]}^{2}} \end{array}$$

### 3.2. Approximations of the Displacement Field

By adopting the NURBS basis functions, the approximate mechanical displacement field u(ξ,η) of the FG-GRP plates based on RPT can be described as
(25)$$u(\xi ,\eta )={\left(\begin{array}{cccc} u_{0} & v_{0} & w_{b} & w_{s} \end{array}\right)}^{T}=\sum_{I}^{m\times n}R_{I}^{P}(\xi ,\eta )d_{I}$$
in which m×n represents the total number of basic functions, and $d_{I}={\left[\begin{array}{cccc} u_{0I} & v_{0I} & w_{bI} & w_{sI} \end{array}\right]}^{T}$ denotes the displacement vector of control point *P_I_*.

Taking Equation (25) into Equation (9) yields
(26)$${\left[\begin{array}{cccc} \varepsilon_{m}^{T} & \kappa_{b}^{T} & \kappa_{s}^{T} & \varepsilon_{s}^{T} \end{array}\right]}^{T}=\sum_{I=1}^{m\times n}{\left[\begin{array}{cccc} {\left(B_{I}^{M}\right)}^{T} & {\left(B_{I}^{b1}\right)}^{T} & {\left(B_{I}^{b2}\right)}^{T} & {\left(B_{I}^{s}\right)}^{T} \end{array}\right]}^{T}d_{I}$$
where
(27)$$\begin{array}{l} B_{I}^{M}=\left[\begin{array}{cccc} R_{I,x} & 0 & 0 & 0\\ 0 & R_{I,y} & 0 & 0\\ R_{I,y} & R_{I,x} & 0 & 0 \end{array}\right]\\ B_{I}^{b1}=-\;\left[\begin{array}{cccc} 0 & 0 & R_{I,xx} & 0\\ 0 & 0 & R_{I,yy} & 0\\ 0 & 0 & 2R_{I,xy} & 0 \end{array}\right]\\ B_{I}^{b2}=\left[\begin{array}{cccc} 0 & 0 & 0 & R_{I,xx}\\ 0 & 0 & 0 & R_{I,yy}\\ 0 & 0 & 0 & 2R_{I,xy} \end{array}\right]\\ B_{I}^{s}=\left[\begin{array}{cccc} 0 & 0 & 0 & R_{I,x}\\ 0 & 0 & 0 & R_{I,y} \end{array}\right] \end{array}$$

### 3.3. Approximations of the Electric Potential

After approximating the potential field, the system can be discretized into several sublayer elements along the thickness direction [51]. The approximate potential field $\phi^{i}(z)$ is written as follows:(28)$$\phi^{i}(z)=R_{\phi }^{i}\phi^{i}$$
in which $\phi^{i}$ represents electric potential of the upper and lower layers related to the *i*th sublayer,
(29)$$\phi^{i}=[\begin{array}{cc} \phi^{i-1} & \phi^{i} \end{array}],(i=1,2,\ldots ,n_{sub})$$

Here, assuming that each individual piezoelectric layer has the same potential at the same height, we may obtain the electric field of each sublayer unit by substituting Equation (28) into Equation (13).
(30)$$E^{i}=-\nabla R_{\phi }^{i}\phi^{i}=-B_{\phi }^{i}\phi^{i}$$
where
(31)$$B_{\phi }^{i}={\left\{\begin{array}{ccc} 0 & 0 & R_{I}/h \end{array}\right\}}^{T}$$

### 3.4. Isogeometric FG-GRP Plate Formulation

The governing equation of motion of the FG-GRP plate can be found by inserting Equations (12), (26) and (30) into Equation (18):(32)$$\left[\begin{array}{cc} M_{uu}^{e} & 0\\ 0 & 0 \end{array}\right]\left[\begin{array}{c} \ddot{d}\\ \ddot{\phi } \end{array}\right]+\left[\begin{array}{cc} K_{uu}^{e} & K_{u\phi }^{e}\\ K_{\phi u}^{e} & -K_{\phi \phi }^{u} \end{array}\right]\left[\begin{array}{c} d\\ \phi \end{array}\right]=\left[\begin{array}{c} f_{e}\\ Q_{e} \end{array}\right]$$
where
(33)$$M_{uu}^{e}=\int_{\Omega }\Lambda^{T}m\Lambda d\Omega$$
(34)$$\begin{array}{l} K_{uu}^{e}=\int_{\Omega }B_{u}^{T}cB_{u}d\Omega ,\;K_{\phi \phi }^{e}=\int_{\Omega }B_{\phi }^{T}gB_{\phi }d\Omega \\ K_{u\phi }^{e}=\int_{\Omega }B_{u}^{T}{\tilde{e}}^{T}B_{\phi }d\Omega ,\;K_{\phi u}^{e}={\left(K_{u\phi }^{e}\right)}^{T} \end{array}$$
(35)$$f_{e}=\int_{\Omega }f_{s}\bar{\Lambda }d\Omega ,\;Q_{e}=\int_{\Omega }R_{I}^{T}q_{E}d\Omega$$
where
(36)$$B_{u}={\left[\begin{array}{ccc} B_{I}^{M} & B_{I}^{b1} & \begin{array}{cc} B_{I}^{b2} & B_{I}^{s} \end{array} \end{array}\right]}^{T}$$
(37)$$c=\left[\begin{array}{cc} D_{m} & 0\\ 0 & D_{s} \end{array}\right]$$
(38)$$\tilde{e}=[\begin{array}{ccccc} e_{m}^{T} & ze_{m}^{T} & z^{3}e_{m}^{T} & e_{s}^{T} & z^{2}e_{s}^{T} \end{array}]$$
(39)$$\begin{array}{l} e_{m}=\left[\begin{array}{ccc} 0 & 0 & 0\\ 0 & 0 & 0\\ e_{31} & e_{32} & 0 \end{array}\right]\\ e_{s}=\left[\begin{array}{cc} 0 & e_{15}\\ e_{15} & 0\\ 0 & 0 \end{array}\right] \end{array}$$
(40)$$g=\left[\begin{array}{ccc} \kappa_{11} & 0 & 0\\ 0 & \kappa_{22} & 0\\ 0 & 0 & \kappa_{33} \end{array}\right]$$
(41)$$m=\left[\begin{array}{ccc} I_{0} & 0 & 0\\ 0 & I_{0} & 0\\ 0 & 0 & I_{0} \end{array}\right],\;I_{0}=\left[\begin{array}{ccc} I_{1} & I_{2} & I_{4}\\ I_{2} & I_{3} & I_{5}\\ I_{4} & I_{5} & I_{6} \end{array}\right]$$
(42)$${\left(I_{1},I_{2},I_{3},I_{4},I_{5},I_{6}\right)}^{\left(k\right)}=\int_{-h/2}^{h/2}\left(1,z,z^{2},f\left(z\right),zf\left(z\right),f^{2}\left(z\right)\right)\rho_{k}dz$$
(43)$$\Lambda =\left[\begin{array}{cc} \Lambda_{1} & \begin{array}{cc} \Lambda_{2} & \Lambda_{3} \end{array} \end{array}\right]$$
(44)$$\begin{array}{l} \Lambda_{1}=\left[\begin{array}{cccc} R_{I} & 0 & 0 & 0\\ 0 & 0 & -R_{I,x} & 0\\ 0 & 0 & 0 & R_{I,x} \end{array}\right]\\ \Lambda_{2}=\left[\begin{array}{cccc} 0 & R_{I} & 0 & 0\\ 0 & 0 & -R_{I,y} & 0\\ 0 & 0 & 0 & R_{I,y} \end{array}\right]\\ \Lambda_{3}=\left[\begin{array}{cccc} 0 & 0 & R_{I} & R_{I}\\ 0 & 0 & 0 & 0\\ 0 & 0 & 0 & 0 \end{array}\right] \end{array}$$
(45)$$\bar{\Lambda }={\left\{\begin{array}{cccc} 0 & 0 & R_{I} & R_{I} \end{array}\right\}}^{T}$$
where $f_{s}$ denotes the transverse mechanical surface load, $q_{E}$ is charge surface density, and
(46)$$q_{E}=k_{33}\frac{U}{h}$$
where *U* is the applied voltage to the upper and lower surfaces.

The element stiffness matrix for buckling analysis can be represented as:(47)$$K_{g}^{e}=\int_{\Omega }B_{g}^{T}\tau B_{g}d\Omega$$
where
(48)$$B_{I}^{g}=\left[\begin{array}{cccc} 0 & 0 & R_{I,x} & R_{I,x}\\ 0 & 0 & R_{I,y} & R_{I,y} \end{array}\right]$$
(49)$$\tau =\left[\begin{array}{cc} \sigma_{x}^{0} & \tau_{xy}^{0}\\ \tau_{xy}^{0} & \sigma_{y}^{0} \end{array}\right]$$

The edges of the FG-GRP plate are subjected to a uniaxial load and biaxial loads, as shown in Figure 3a,b. The connection between buckling forces and load parameter index is expressed as follows:(50)$$\sigma_{x}^{0}=\xi_{1}P_{0},\;\sigma_{y}^{0}=\xi_{2}P_{0},\;\tau_{xy}^{0}=0$$
where $\xi_{1}$ and $\xi_{2}$ are parameters that control load conditions.

To evaluate the impact of the Winkler elastic foundation, the additional matrix $K_{f}^{e}$ of the elastic foundation is introduced:(51)$$K_{f}^{e}=\int_{\Omega }k_{f}N^{T}Nd\Omega$$
where $N=\left[\begin{array}{cccc} R_{1}^{P}I_{4\times 4} & R_{2}^{P}I_{4\times 4} & \ldots & R_{N}^{P}I_{4\times 4} \end{array}\right]$, $k_{f}$ is the elastic foundation coefficient.

Eliminating the potential and assembling elements, the governing equations of FG-GRP are represented as:(52)$$\begin{array}{l} M_{uu}\ddot{d}+(K_{uu}+K_{u\phi }K_{\phi \phi }^{-1}K_{\phi u})d=f+K_{u\phi }K_{\phi \phi }^{-1}Q\\ \Leftrightarrow M\ddot{d}+Kd=F \end{array}$$

The governing equations for static analysis, free vibration and buckling behaviors are obtained as follows
(53)$$\left(K+K_{f}\right)U=F$$
(54)$$\left(K+K_{f}-w^{2}M\right)d=0$$
(55)$$\left(K+K_{f}-\lambda_{cr}K_{g}\right)d=0$$
where
(56)$$M=M_{uu}$$
(57)$$K=K_{uu}+K_{u\phi }K_{\phi \phi }^{-1}K_{\phi u}$$
(58)$$F=f+K_{u\phi }K_{\phi \phi }^{-1}Q$$

## 4. Numerical Investigation

In this section, static analysis, free vibration and buckling behaviors of FG-GRP plates on Winkler elastic foundation are investigated by isogeometric analysis. Unless otherwise specified, an FG-GRP plate with geometric size a=b=0.1 m, thickness h=0.01 m is considered. Additionally, the graphene volume fraction is set as 0.5%. The GPLs’ piezoelectric characteristics can be defined as $e_{31,G}=\alpha e_{31,M}$, $e_{32,G}=\alpha e_{32,M}$, $e_{24,G}=\alpha e_{24,M}$, $e_{15,G}=\alpha e_{15,M}$, $\kappa_{11,G}=\alpha \kappa_{11,M}$, $\kappa_{22,G}=\alpha \kappa_{22,M}$ and $\kappa_{33,G}=\alpha \kappa_{33,M}$, where α is named as piezoelectricity multiple [30]. Furthermore, the relevant parameters of GPL and PVDF [28] are:$$E_{gpl}=1010\;\mathrm{GPa},\;v_{gpl}=0.186,\;\rho_{gpl}=1062.5\;{\mathrm{kg}/m}^{3}\;a_{gpl}=2.5\;\mu m,\;b_{gpl}=1.5\;\mu m,\;h_{gpl}=1.5\;\mathrm{nm}\;E_{M}=1.44\;\mathrm{GPa},\;v_{M}=0.290,\;\rho_{M}=1920.0\;{\mathrm{kg}/m}^{3}\;e_{31,M}=50.535\times 10^{-3}\;{C/m}^{2},\;e_{32,M}=13.212\times 10^{-3}\;{C/m}^{2},\;e_{24,M}=12.65\times 10^{-3}\;{C/m}^{2},\;e_{15,M}=15.93\times 10^{-3}\;{C/m}^{2}\;\kappa_{11,M}=0.5385\times 10^{-9}\;C/\mathrm{Vm},\;\kappa_{22,M}=0.6638\times 10^{-9}\;C/\mathrm{Vm},\;\kappa_{33,M}=0.5957\times 10^{-9}\;C/\mathrm{Vm}$$

The following dimensionless formula is used in this paper unless otherwise specified:
$$\begin{array}{c} w_{r}=100^{*}wE_{M}h^{3}/\left(12^{*}q_{0}a^{4}\left(1-v_{M}^{2}\right)\right)\\ \omega_{l}=\omega h\sqrt{\left(1-v_{M}^{2}\right)\rho_{M}/E_{M}}\\ P_{cr}^{*}=\left(P_{cr}a^{2}\right)/\left(E_{M}h^{3}1000\right) \end{array}$$


Firstly, the convergence of the IGA technique is examined by adopting (p+1)(q+1) Gauss quadrature, where p and q indicate the orders of the basic functions along the directions of ξ and η, respectively. Furthermore, the effects of the geometric parameters of the plate, graphene volume fraction, the distribution pattern, and the Winkler elastic coefficient on the deflections, frequencies and bucking loads of FG-GRP plates resting in Winkler elastic foundation are discussed.

### 4.1. Verification

Firstly, the FG-GRP plates are modelled with 7 × 7, 11 × 11 and 19 × 19 control points as shown in Figure 4a–c. Then, the convergence and accuracy of the present solutions at different mesh levels are presented in Table 1 for an FG-GRP plate under CCCC and SSSS boundary conditions. Table 1 displays the dimensionless frequencies and critical buckling loads of the FG-GRP plates. It is observed from Table 1 that the present calculation results are highly consistent with those listed in studies [30,31], and the IGA method has reached the convergence state when the control points are greater than 11 × 11. As a result, the number of control points utilized in the following examples in this paper is 11 × 11. Lastly, the effects of Winkler elastic coefficient $k_{l}$ on the central deflection are shown in Table 2. It is seen that the normalized deflections in this paper match well with the results of Shojaee et al. [52]. Table 3 compares the dimensionless vibration frequency and percentage frequency change of the FG-GRP plate to the calculation results of Mao et al. [30] for different GPL distribution patterns. As shown in Table 3, the percentage change on frequency in terms of the X type is the highest one among three kinds of distribution patterns. Table 4 compares the dimensionless critical buckling loads $P_{cr}^{*}=(P_{cr}a^{2})/(E_{m}h^{3})$ of FG-GRP plates to the calculation findings of Mao et al. [31]. It can be seen that the calculation results are mostly consistent.

### 4.2. Static Analysis

Figure 5a,b show the effect of the number of layers on the percentage deflection change of the FG-GRP plate under different graphene distribution patterns. For the U type, the number of layers has no effect on the percentage deflection change. Moreover, the influence of the layers on the percentage deflection change is extremely obvious for the X type and the O type. For the X type, with a rise in the number of layers *N*, more GPL reinforcements are spread on both sides of the plate to enhance the stiffness of the system. While for the other one, the O type plays opposite roles on the stiffness and dimensionless central deflections. Simultaneously, with increasing the number of layers *N*, the percentage deflection changes corresponding to the X and O types gradually converge when N≥16. Therefore, in the following analysis, the number of layers N=16 is employed for computation to improve calculation efficiency.

The effective material properties for the composite are determined by the length and thickness of the graphene reinforcements but have nothing to do with the width in Halpin Tsai’s parallel model [28]. Assuming $b_{gpl}=1.5\;\mu m$, Figure 6a–d illustrate the percentage change on deflection under a different graphene length-to-thickness ratio $a_{gpl}/h_{gpl}$. It is clearly discerned from this figure that the percentage deflection change increases with the increase in $a_{gpl}/h_{gpl}$, indicating that smaller and thinner graphene reinforcements can more effectively enhance the stiffness of the FG-GRP plate; this is due to the fact that the greater the contact area between the graphene platelet and the matrix material, the better the bonding ability. Furthermore, compared to Figure 6a–d, it can be shown that the percentage deflection change for a given pattern under different boundary conditions is almost the same. As a result, the following analysis only takes into account the percentage deflection changes under the SSSS boundary condition.

The impact of the Winkler elastic coefficients $k_{l}$ on the normalized central deflections $w_{r}$ of the FG-GRP plate under varied graphene volume fractions $V_{gpl}$ is depicted in Figure 7a. The dimensionless central deflections of the FG-GRP plate decrease as the Winkler elastic coefficients increase, as seen in this figure. Simultaneously, it is discerned that the normalized central deflections decrease with increasing GPL volume fractions. This is due to the fact that graphene reinforcements can effectively enhance the system stiffness, and the higher the graphene volume fraction (less than 1%), the stronger the impact. Figure 7b displays the effects of the Winkler elastic coefficient $k_{l}$ on the dimensionless central deflections $w_{r}$ under different graphene distribution patterns. It can be observed that under different graphene distribution patterns, the decreasing central deflections can be caused by the increasing elastic coefficients.

### 4.3. Vibration Analysis

In what follows, the influence of detailed parameters on free vibration analysis for an FG-GRP plate with $b_{gpl}=1.5\;\mu m$ is analyzed. Figure 8 illustrates the percentage change on frequency under a different graphene length-to-thickness ratio $a_{gpl}/h_{gpl}$, graphene volume fraction and graphene distribution patterns (the U type, the X type and the O type). It can be clearly seen from this figure that the percentage frequency change increases with the increase in $a_{gpl}/h_{gpl}$, indicating that an increasing graphene length-to-thickness ratio can enhance the stiffness of the FG-GRP plate. Furthermore, this figure demonstrates that, regardless of graphene distribution patterns, the percentage change on frequency increases with an increasing graphene volume fraction.

As shown in Figure 9, the effect of plate geometric dimension on the free vibration characteristics is discussed, including aspect ratio a/b and length-to-thickness ratio a/h. The FG-GRP plate with a=0.1 m is applied in the following parametric studies. Because of the uniformity, the geometry dimension of the plate has no effect on the vibration characteristics corresponding to the U type. Since the surface area of the square plate in this paper is larger than that of the rectangular plate, the GPL reinforcements of the X type are more distributed at the top and bottom surfaces of the square plate when the same GPL reinforcements are used. As a result, the square plate presents more remarkable reinforcement effects than the rectangle plate in terms of the X-type plate.

The effect of the Winkler elastic coefficient on the dimensionless vibration frequencies of the FG-GRP plate is shown in Figure 10a,b. Figure 10a presents the influences of the Winkler elastic coefficient $k_{l}$ on the dimensionless vibration frequency $\omega_{l}$ under different graphene volume fractions $V_{gpl}$. It is observed that increasing Winkler elastic coefficients leads to increased dimensionless vibration frequencies. Figure 10b examines the dimensionless vibration frequency of the FG-GRP plate versus elastic coefficients under different graphene distribution patterns. It is clearly demonstrated that the increasing frequencies increase with increasing elastic coefficients under different graphene distribution patterns, which shows that the increase in Winkler elastic coefficient can effectively enhance the stiffness of the FG-GRP plate.

### 4.4. Buckling Analysis

To explore the impact of detailed parameters on the buckling behaviors, the uniaxial or biaxial in-plane forces must be applied to the piezoelectric plates.

Figure 11a–d show the effect of graphene volume fraction $V_{gpl}$ on the normalized critical buckling load $P_{cr}^{*}$ of the FG-GRP plates for different boundary conditions and graphene distribution patterns. The normalized critical buckling load $P_{cr}^{*}$ subjected to uniaxial load is approximately twice as large as that subjected to biaxial load, owing to the fact that the biaxial load in *x* and *y* directions is loaded from four directions, limiting the load it can carry. Next, the normalized critical buckling load $P_{cr}^{*}$ is the largest one under CCCC, followed by CCSS, CSSS, and SSSS boundary conditions. For the uniaxial and biaxial loads, the normalized critical buckling load $P_{cr}^{*}$ increases with the increasing graphene volume fraction $V_{gpl}$. When the graphene volume fraction is the same, the X-type plate has the highest critical buckling load, while the O-type plate has the lowest one. Hence, the following study exclusively analyzes the buckling behavior of the X-type plate under the SSSS boundary condition for the sake of brevity.

The impact of the length-to-width ratio a/b and the graphene length-to-thickness ratio $a_{gpl}/h_{gpl}$ on the normalized critical buckling load of the X-type plate under uniaxial load is examined in Figure 12. When $a_{gpl}/h_{gpl}$ increases from 0 to 2, the normalized critical buckling load $P_{cr}^{*}$ increases significantly, and when $a_{gpl}/h_{gpl}$ continues to rise, $P_{cr}^{*}$ rises slowly, indicating that the longer and thinner GPL reinforcements can reinforce the buckling resistance. Furthermore, under certain $a_{gpl}/h_{gpl}$, the normalized critical buckling load $P_{cr}^{*}$ increases with length-to-width ratio a/b increasing. Under two loading conditions, Figure 13 shows the effect of the Winkler elastic foundation’s elastic coefficient $k_{l}$ on the normalized critical buckling load $P_{cr}^{*}$ of the X type. The normalized critical buckling load increases as the elastic coefficient grows, showing that increasing the elastic coefficient enhances system stiffness.

## 5. Conclusions

Isogeometric analysis integrated with the refined plate theory is utilized in this paper to analyze the static bending, free vibration and buckling behaviors of functionally graded graphene-platelet-reinforced piezoelectric plates on Winkler elastic foundations. When Hamilton’s principle is combined with isogeometric analysis, the isogeometric finite element governing equations of the functionally graded graphene-platelet-reinforced piezoelectric plate are obtained on the basis of the refined plate theory. Comparisons with other studies validate the accuracy and convergence of the present isogeometric analysis. Furthermore, the influences of a plate’s geometrical parameters, the elastic coefficient, and GPLs’ physical and geometric properties on the percentage change of central deflections, frequencies and critical buckling loads of a functionally graded graphene-platelet-reinforced piezoelectric plate are thoroughly investigated under different boundary conditions and graphene distribution patterns.

The results of this paper indicate that adding a small amount of GPL not only decreases the central deflections, but also increases the vibration frequencies and critical buckling loads of functionally graded graphene-platelet-reinforced piezoelectric plates. Additionally, the X type represents the optimum distribution form, i.e., the higher the graphene volume fractions spread into the surface layer, the more effective it is to enhance the stiffness of the system. Additionally, increasing the graphene length-to-thickness ratio can significantly enhance system stiffness, that is, the longer and thinner the GPLs are, the more obvious the reinforcement is. In terms of a plate’s geometrical parameters, the effect of a functionally graded graphene-platelet-reinforced piezoelectric plate’s aspect ratio on central deflections, vibration frequencies and critical buckling loads should be handled differently depending on the graphene distribution pattern. For the X type, increasing the aspect ratio or plate surface area can improve system stiffness. In terms of the impacts of boundary conditions on the system, the stiffness of the functionally graded graphene-platelet-reinforced piezoelectric plate under the CCCC boundary condition is the highest one, followed by CCSS, CSSS, and SSSS. Furthermore, the dimensionless critical buckling load subjected to the uniaxial force is approximately twice as large as that subjected to the biaxial force. Effectively increasing the elastic coefficient of the Winkler elastic foundation can obtain a remarkable reinforcement effect on the plate stiffness.

## Figures and Tables

**Figure 1 materials-15-05727-f001:**
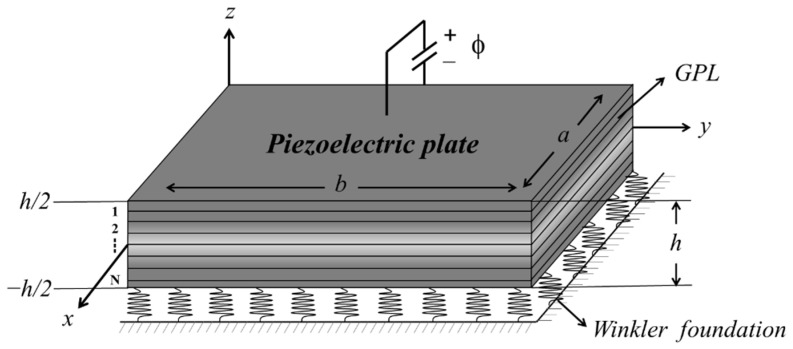
AN FG-GRP plate resting on the Winkler elastic foundation is shown.

**Figure 2 materials-15-05727-f002:**
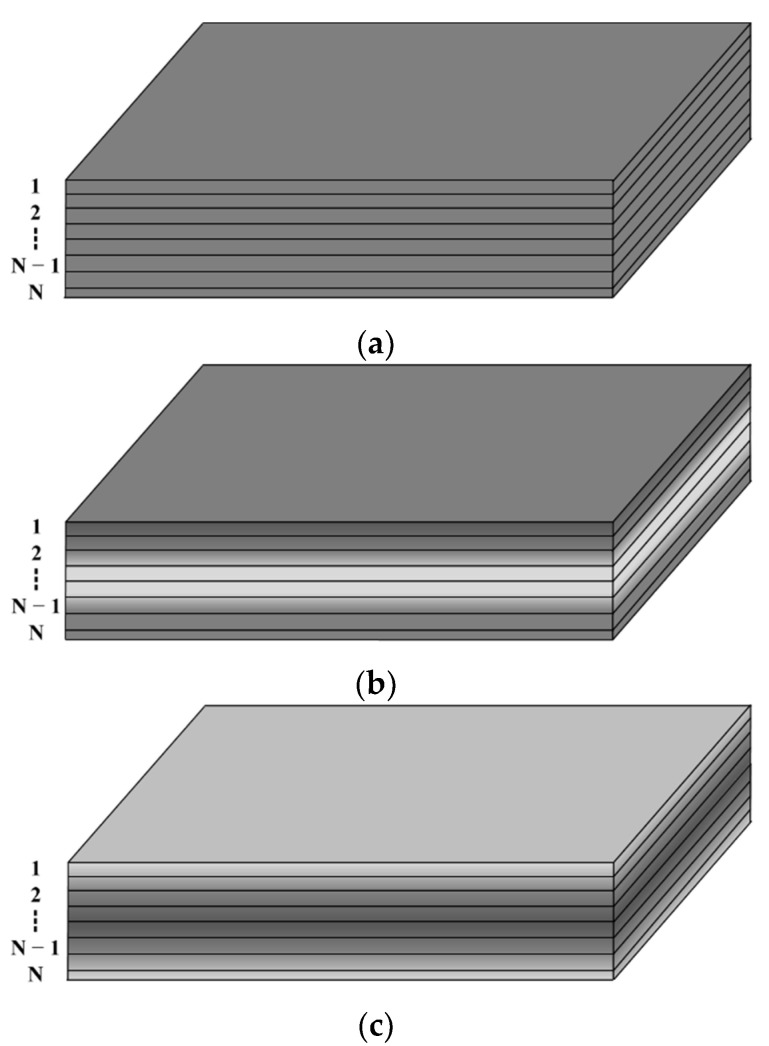
Distribution patterns along the thickness direction of GPLs: (**a**) U type, (**b**) X type, (**c**) O type.

**Figure 3 materials-15-05727-f003:**
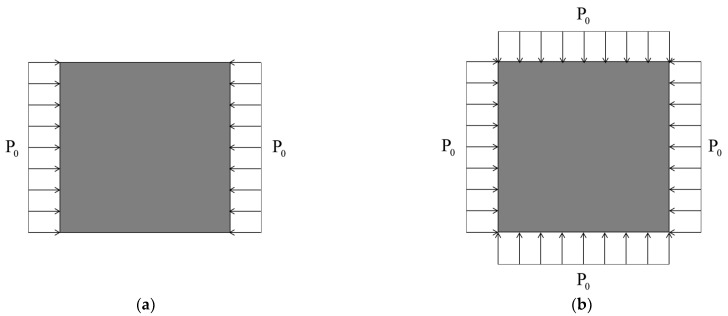
A square plate subjected to: (**a**) uniaxial load; (**b**) biaxial load.

**Figure 4 materials-15-05727-f004:**
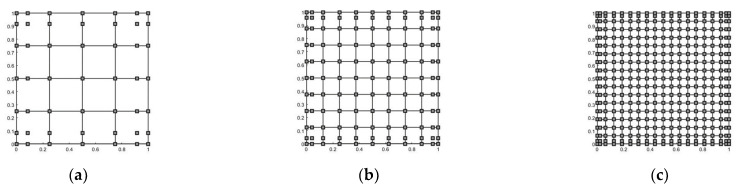
Three meshes of (**a**) 7 × 7 control points, (**b**) 11 × 11 control points, (**c**) 19 × 19 control points.

**Figure 5 materials-15-05727-f005:**
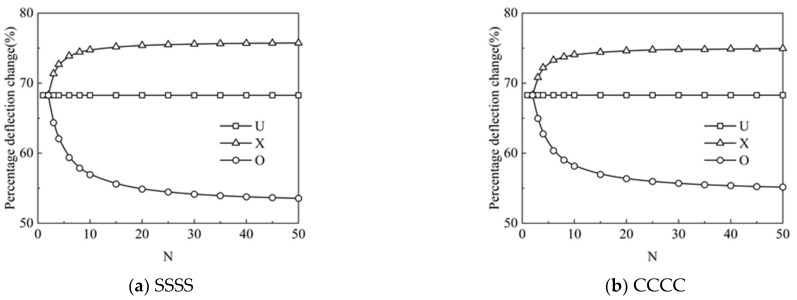
Variation of the percentage deflection change along the total number of layers for the (**a**) SSSS and (**b**) CCCC FG-GRP plates.

**Figure 6 materials-15-05727-f006:**
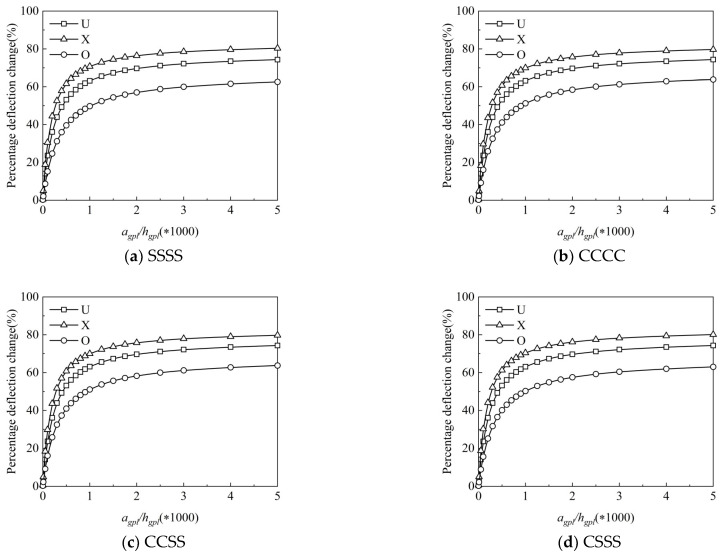
Variation of the percentage deflection change along the graphene length-to-thickness ratio for the (**a**) SSSS, (**b**) CCCC, (**c**) CCSS and (**d**) CSSS FG-GRP plates.

**Figure 7 materials-15-05727-f007:**
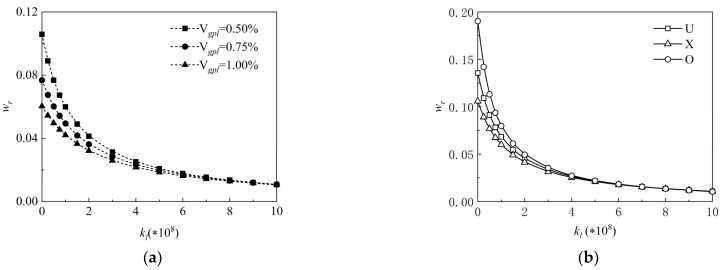
Variation of dimensionless central deflection along the elastic coefficient of the Winkler elastic foundation for an SSSS FG-GRP plate under different (**a**) graphene volume fractions and (**b**) graphene distribution patterns.

**Figure 8 materials-15-05727-f008:**
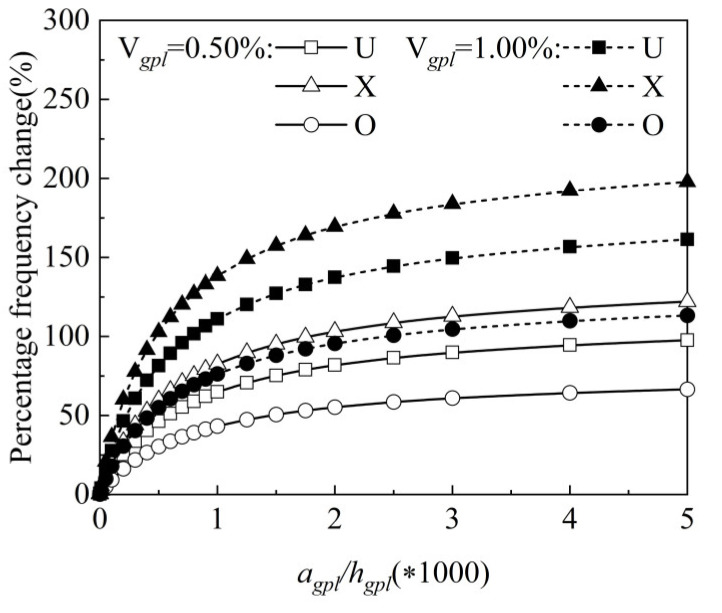
Variation of the percentage frequency change along the graphene length-to-thickness ratio for an SSSS FG-GRP plate.

**Figure 9 materials-15-05727-f009:**
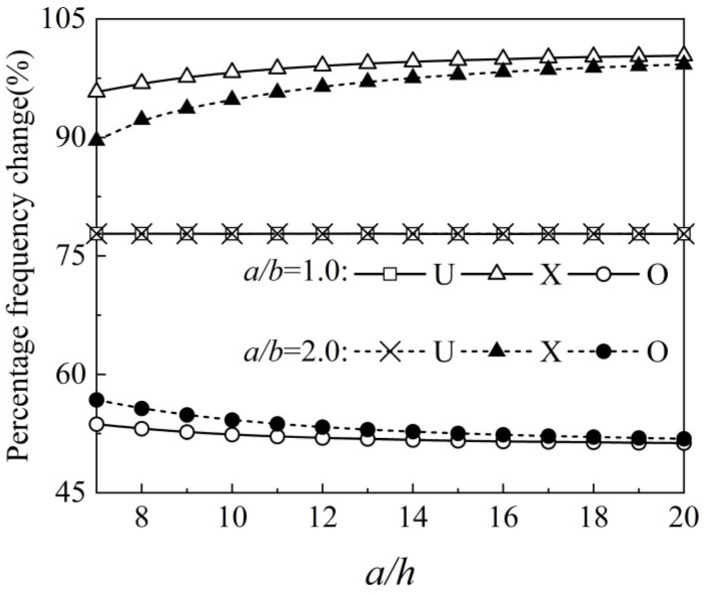
Variation of the percentage frequency change along the side-to-width ratio and side-to-thickness ratio for an SSSS FG-GRP plate.

**Figure 10 materials-15-05727-f010:**
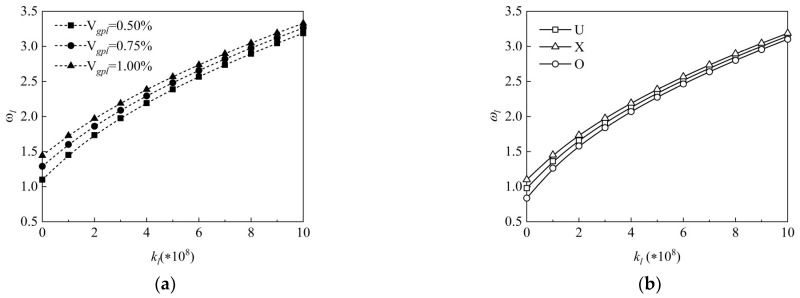
Variation of the dimensionless frequency along the elastic coefficient of the Winkler elastic foundation for an SSSS G-GRP plate under different (**a**) graphene volume fractions and (**b**) graphene distribution patterns.

**Figure 11 materials-15-05727-f011:**
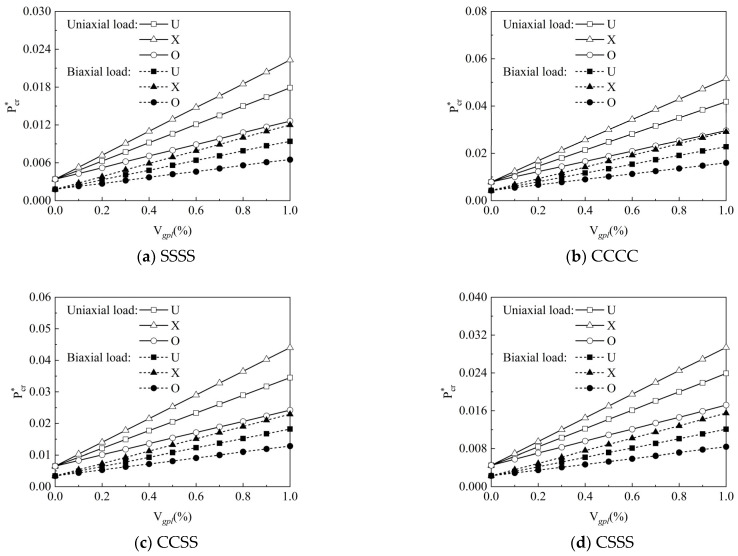
Variation of dimensionless critical buckling load along graphene volume fractions for the (**a**) SSSS, (**b**) CCCC, (**c**) CCSS, and (**d**) CSSS FG-GRP plates subjected to uniaxial and biaxial in-plane forces.

**Figure 12 materials-15-05727-f012:**
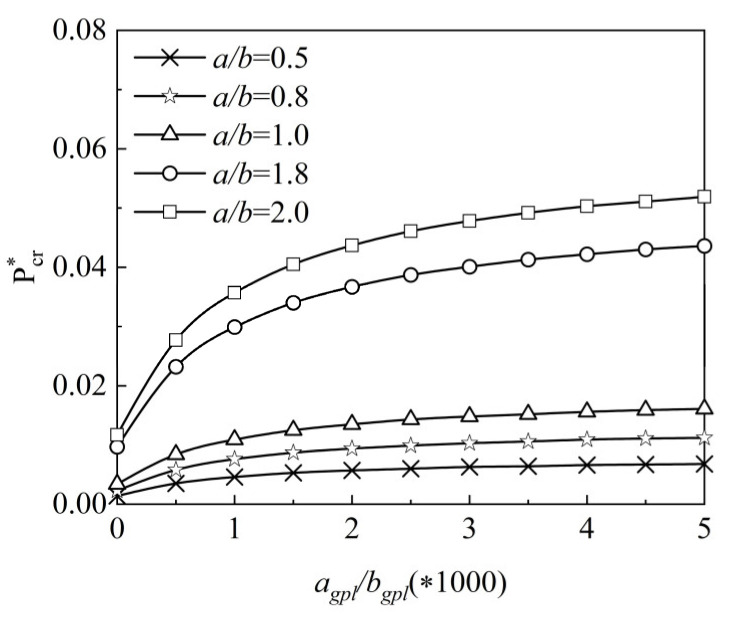
Variation of dimensionless critical buckling load along the graphene length-to-thickness ratio and the side-to-width ratio *a/b*.

**Figure 13 materials-15-05727-f013:**
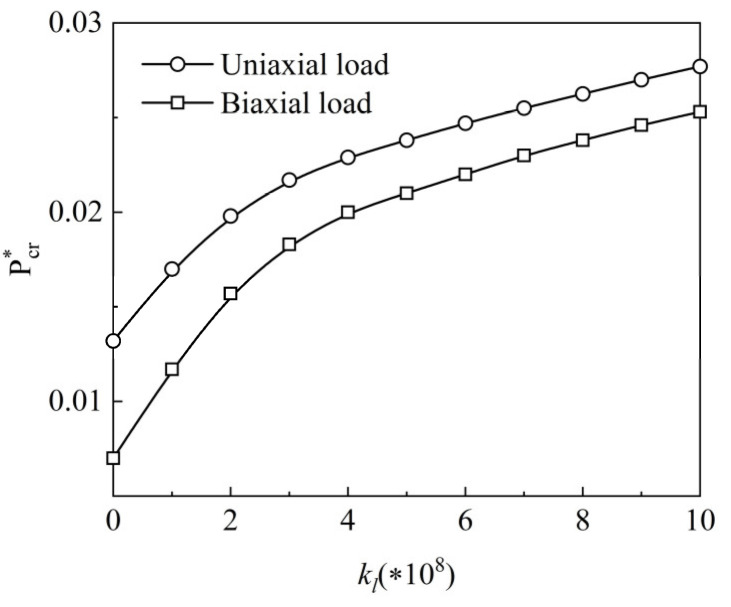
Variation of dimensionless critical buckling load along the elastic coefficient of the Winkler elastic foundation for an SSSS FG-GRP plate.

**Table 1 materials-15-05727-t001:** Convergence of dimensionless frequency and dimensionless critical buckling load of the FG-GRP plates with different control meshes.

BC	Method	ω		$P_{\mathrm{cr}}^{*}$	
		Mode 1	Mode 2	Uniaxial	Biaxial
SSSS	IGA (7 × 7)	0.9777	2.552	0.0097	0.0054
	IGA (11 × 11)	0.9789	2.337	0.0106	0.0056
	IGA (19 × 19)	0.9789	2.337	0.0106	0.0056
	Mao et al. [30,31]	0.9789	2.337	0.0106	0.0056
CCCC	IGA (7 × 7)	1.6728	3.2265	0.0246	0.0133
	IGA (11 × 11)	1.6714	3.1905	0.0248	0.0135
	IGA (19 × 19)	1.6714	3.1905	0.0248	0.0135
	Mao et al. [30,31]	1.6716	3.1906	0.0248	0.0135

**Table 2 materials-15-05727-t002:** Comparison of dimensionless central deflection of square plate on elastic foundation with different boundary conditions.

*k* _l_		BC		
		SSSS	SCSC	CSSS
0	Shojaee et al. [52]	0.0040624	0.0019172	0.0028001
	FSDT	0.0040665	0.0019205	0.0028000
	RPT	0.0040626	0.0019172	0.0027971
5	Shojaee et al. [52]	0.0040097	0.0019053	0.0027602
	FSDT	0.0040254	0.0019116	0.0027804
	RPT	0.0040215	0.0019083	0.0027775
100	Shojaee et al. [52]	0.0032137	0.0017050	0.0023522
	FSDT	0.0033740	0.0017565	0.0024524
	RPT	0.0033707	0.0017533	0.0024498

**Table 3 materials-15-05727-t003:** Dimensionless frequency of the FG-GRP plates with different graphene volume fraction and different distribution patterns under different boundary conditions.

BC	GPL Patterns		V_*gpl*_ (%)			
			0	0.25	0.5	0.75
SSSS	U	Mao et al. [30]	0.551	0.794	0.9791	1.135
		FSDT	0.551	0.7937	0.9789	1.1349
		RPT	0.551	0.7937	0.9789	1.1349
				44.2%	77.8%	106.1%
	X	Mao et al. [30]	0.551	0.8678	1.0974	1.2875
		FSDT	0.5506	0.8648	1.0914	1.279
		RPT	0.5506	0.8648	1.0914	1.279
				57.1%	98.2%	132.3%
	O	Mao et al. [30]	0.551	0.7108	0.8407	0.9534
		FSDT	0.5506	0.7099	0.8389	0.9508
		RPT	0.5506	0.7099	0.8389	0.9508
				28.9%	52.4%	72.7%
CCCC	U	Mao et al. [30]	0.9405	1.3554	1.6716	1.938
		FSDT	0.94	1.3552	1.6714	1.9379
		RPT	0.94	1.3552	1.6714	1.9379
				44.2%	77.8%	106.2%
	X	Mao et al. [30]	0.9405	1.4653	1.8468	2.1632
		FSDT	0.94	1.4669	1.8494	2.1666
		RPT	0.94	1.4669	1.8494	2.1666
				56.1%	96.7%	130.5%
	O	Mao et al. [30]	0.9405	1.2257	1.4576	1.6562
		FSDT	0.94	1.2243	1.4523	1.6493
		RPT	0.94	1.2243	1.4523	1.6493
				30.2%	54.5%	75.5%

**Table 4 materials-15-05727-t004:** Dimensionless critical bucking load of an SSSS FG-GRP plate with different graphene volume fractions and different distribution patterns under uniaxial and biaxial loads.

Load Type	GPL Patterns		V_*gpl*_ (%)			
			0	0.2	0.6	0.8
Uniaxial	U	Mao et al. [31]	0.0034	0.0063	0.0122	0.0151
		FSDT	0.0034	0.0063	0.0122	0.0151
		RPT	0.0034	0.0063	0.0121	0.0150
				185.3%	355.9%	441.2%
	X	Mao et al. [31]	0.0034	0.0073	0.0151	0.0190
		FSDT	0.0034	0.0073	0.0151	0.0190
		RPT	0.0034	0.0072	0.0148	0.0185
				211.8%	435.3%	544.1%
	O	Mao et al. [31]	0.0034	0.0053	0.0089	0.0108
		FSDT	0.0034	0.0053	0.0089	0.0108
		RPT	0.0034	0.0052	0.0089	0.0108
				152.9%	261.8%	317.6%
Biaxial	U	Mao et al. [31]	0.0018	0.0033	0.0063	0.0079
		FSDT	0.0018	0.0033	0.0063	0.0079
		RPT	0.0018	0.0033	0.0064	0.0079
				183.3%	355.6%	438.9%
	X	Mao et al. [31]	0.0018	0.0039	0.0080	0.0101
		FSDT	0.0018	0.0039	0.0080	0.0101
		RPT	0.0018	0.0038	0.0079	0.0100
				211.1%	438.9%	555.6%
	O	Mao et al. [31]	0.0018	0.0027	0.0046	0.0055
		FSDT	0.0018	0.0027	0.0046	0.0055
		RPT	0.0018	0.0027	0.0046	0.0056
				150.0%	255.6%	311.1%

## Data Availability

Not applicable.

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
