# Peer review of "Isogeometric Analysis of Graphene-Reinforced Functionally Gradient Piezoelectric Plates Resting on Winkler Elastic Foundations"

_materials, 2022, doi:10.3390/ma15165727_

Round 1
Reviewer 1 Report
Congratulations for carrying out comprehensive and interesting research. The manuscript is well written.
The contribution made addresses numerical analysis methods for GRFG piezoelectric plates on elastic Winkler foundations. Potential applications of the presented work are many.
The topic is original, of a high level.
As far as I am aware the authors are correct in claiming this fills a gap in the literature rather than simply adding to existing studies.
The approach taken is good and theoretically sound.
The conclusions are consistent with the evidence and arguments
presented and they address the main question posed.
The references are appropriate.
My suggested minor modifications are few and of editorial nature. If the editor agrees, I recommend you simply make necessary adjustments in consultation with the journal production staff. My suggestions are:
- Keep use of acronyms to a minimum because readers might find it difficult to remember them all.
- Lettering in figures is often too small to be read easily. Using a larger font size would be good.
- Shading in some figures is too dark, Figs. 1, 2.
- Quote values to reasonable precision, e.g. five figure precision in Table 1 seems unnecessary.
- Avoid use of acronyms in the Conclusion section.
Reviewer 2 Report
Authors of this study have performed an investigation on the free vibration, bending and also buckling of FG-GPLRC plates resting on an elastic foundation and integrated with piezoelectric layers. The analysis is based on a higher order shear deformation theory and also isogeometric formulation. The developed solution method may be used for arbitrary combinations of boundary conditions and loading type. Results of this study are well-compared with the available data in the open literature and novel numerical results are then provided which may be useful for the future readers. In general this work is interesting and deserves to be published. It is well-organised and proposes a step by step procedure for the future readers. In my opinion it may be accepted in its current form
Reviewer 3 Report
The authors provide a mathematical model to analyze some important properties of graphene-based piezoelectric plates.
The equation and results are well presented, supported with tables and figures, and commented in comparison with previous work. However, the resolution of the figures could be improved, and the caption in Figure 7 should also be revised to refer to (a) and (b).
